# Synthesis and Properties Comparison of Low Dielectric Silicon Containing Polyimides

**DOI:** 10.3390/ma15082755

**Published:** 2022-04-08

**Authors:** Yuwei Chen, Yidong Liu, Yonggang Min

**Affiliations:** Department of Materials and Energy, Guangdong University of Technology, No. 100 Waihuanxi Road, Guangzhou HEMC, Guangzhou 510006, China; polebear1996@outlook.com

**Keywords:** silicon, polyimides, dielectric properties, structural design

## Abstract

Recent studies have shown that the introduction of silicon can effectively improve the dielectric properties of polyimide (PI), and the introduction of a silicon–oxygen bond can increase the flexibility of the PI molecular structure, which is conducive to reducing the moisture absorption rate of PI materials. In this experiment, 1,3-bis(3-aminopropyl)-1,1,3,3-tetramethyl disiloxane (DSX) was mixed with 4,4′-diaminodiphenyl ether (ODA) in different proportions. A series of PI films containing silicon was obtained by random polymerization with pyromellitic dianhydride (PMDA), 3,3′,4,4′-diphenylketotetrahedral anhydride (BTDA) and biphenyl dianhydride (BPDA), and then tetrad copolymerization with three kinds of dianhydrides. At the same time, the PI structures were put into calculation software to obtain the simulated polarization results, and then the films were characterized by various properties. The results showed that the characterization results were consistent with that of simulation, and the best overall PI formula was when the ratio of diamines was 1:9 and mixed with PMDA. The performance data were as follows: the vitrification temperature was about 320 °C, T_5_ was 551 °C, water absorption was 1.56%, dielectric constant (Dk) was 2.35, dielectric loss (Df) was 0.007, tensile strength was 70 MPa and elongation at break was 10.2%.

## 1. Introduction

With the proposal and gradual development of 5G, communication equipment has higher and higher property requirements for materials. Due to the frequency of communication waves rising to the GHz level, the loss in wave propagation caused by using traditional materials as the substrate of antenna arrays is much greater than before. To solve this problem, it is necessary to find new materials with lower Dk and Df or to optimize the dielectric properties of materials through physical and chemical methods [1,2,3,4,5]. PI is widely used in the electronics industry because of its excellent mechanical properties, insulation properties, heat resistance and corrosion resistance [6,7]. At present, the material of the receiving antenna group of many smart terminals, such as mobile phones, is PI. The commercial PI materials at present have Dk of more than 3.0 and Df of 10^−2^, which is far from meeting the requirements of 5G or even higher frequency communication wave frequencies [8,9,10,11,12,13,14].

In recent years, many methods have been put forward to improve the dielectric properties of PI. For chemical modification, fluorine-containing or silicon-containing monomers and original dianhydride diamine monomers can be copolymerized to acquire the polymer with the target structure [15,16]. This is because of the low polarity of corresponding bonds introduced into the polymer molecular system effectively reducing the molecular polarity and the electronic polarization so as to reduce the Dk. Second, compared with the original monomers, the introduced molecule has a larger molecular weight and more complex molecular structure, increasing the asymmetry of the new molecular structure and the unit mole mass, which reduce the proportion of the easily polarized imide groups in the unit structure, thus reducing the polarity of the molecule and reducing the Dk and Df [17,18,19,20]. From the aspect of physical modification, the introduction of a hollow structure is very effective. For example, the doping of nanometer hollow microspheres or hollow particles in PAA glue introduces air of low Dk into the polymer system, thus greatly improving the dielectric properties of PIs. Other researchers used light, heat, radiation and initiators to activate monomer molecules into free radicals, which polymerized to produce porous PI materials [21,22,23,24].

Many teams studying silicon-containing PIs with low permittivity carried out research from the perspective of physical doping, while few cases were studied from the perspective of chemical synthesis, and most of them were commercial patents that were not open to the public.

Based on chemical modification, this study focused on the ways to improve the dielectric properties of PIs from the perspective of molecular synthesis. DSX and ODA were mixed in different proportions and randomly polymerized with PMDA, BTDA and BPDA, respectively, and then tetrad-copolymerized with these three dianhydrides to obtain a series of silicon-containing PIs. Meanwhile, the effects of different monomer combinations on the dielectric properties of PI materials were investigated by simulation calculation and synthesis experiments, and the results were generally consistent. The effects of different monomer combinations and frequencies on the dielectric properties of PIs were investigated, and the monomer formulations with relatively good comprehensive properties were selected. It was confirmed that Dk decreased as the silicon content increased, and BPDA was more suitable than BTDA when introduced into the PI structure to decrease Dk and Df. The best formula was when the ratio of diamines was 1:9 and mixed with PMDA. The performance data were as follows: vitrification temperature was about 320 °C, T5 was 551 °C, water absorption was 1.56%, Dk was 2.35, Df was 0.007, tensile strength was 70 MPa and elongation at break was 10.2%.

## 2. Simulation Calculation

This experiment mainly studied the dielectric properties. According to the relationship between different properties and chemical structure, the Dk and Df were related to the polarizability. Therefore, the polarization was chosen as the target of this simulation calculation, and the molecular results of PI structures polymerized by different monomers were calculated to obtain the magnitude relationship of polarization so as to deduce the changing trend of Dk and its relationship with different monomer structures.

First, we determined that the diamines and dianhydrides involved in the experiment were ODA, DSX, PMDA, s-BPDA and BTDA, respectively, and the PI molecular structures that might be synthesized by ternary or quaternary polymerization of them, as shown in Figure 1 and Figure 2. Then, the molecular structures above were imported into the simulation calculation software, Materials Studio (7.0, Accelrys, San Diego, CA, USA), and ‘Polar’ was calculated. Finally, the polarization results were obtained, as shown in Table 1.

It should be noted that under ideal conditions, all monomers labeled as participating in polymerization were polymerized in the main chain, and only the molecular structure of the main chain covering all kinds of monomers was calculated without considering the difference caused by the number of short chains in polymers and the ratio of monomers. The cross-linking and crystallization during polymerization and imidization were preset (the cross-linking could not be controlled, and crystallization was avoided as far as possible through the experimental heating process and cooling process design).

Since the structures introduced in the simulation were the unit structures of each PI molecule, the volumes of the unit structure containing different amounts of monomers were not equal, so only the structures with the same number of monomers were compared simultaneously. Polarization result 1/Angstrom^3^, polarization result 2/cm^3^ and polarization result 3/a.u. are three physical quantities representing the polarizability from different views. As can be seen in Table 1, the introduction of DSX indeed reduced the polarization of the PI molecular structure. However, the data also showed that if ODA was completely replaced with DSX, the polarization would increase somewhat, indicating that when the silicon content reached a certain level, the polarization would decrease adversely. For the polymerization containing DSX, the polarizability of PI films only polymerized with PMDA was lower than that with extra BPDA and BTDA. For the tetrad polymerization containing DSX, the polarizability of the PI film containing PMDA and BPDA was lower than that containing PMDA and BTDA. Therefore, according to the simulation results, the PI films synthesized by ODA/DSX//PMDA and ODA/DSX//BPDA/PMDA had the best dielectric properties in tetrad polymerization.

## 3. Experiment

### 3.1. Reagents

PMDA (99%), DSX (97%), ODA (99.5%), N,N-Dimethylacetamide (DMAc), BPDA (97%) and BTDA (99%) were bought from Macklin (Shanghai, China).

### 3.2. Ternary Polymerization of PMDA, ODA and DSX

In this experiment, the molar ratio of PMDA to diamines was 1:1, and the molar ratios of DSX to ODA were 1:3, 1:5, 1:7, 1:9 to 1:17, respectively, and the solvent was DMAc. After complete dissolution of DMAc and diamine, PMDA was added evenly and slowly for homopolymerization. The experimental conditions were as follows: the temperature was controlled at 24–26 °C, the ambient humidity was about 35%, the PMDA feeding time was controlled within 30–40 min and the stirring speed was constant during feeding. After feeding, the speed was adjusted according to the viscosity of the glue, then continued to react for 6 h, and stirring was stopped. Finally, a series of silicon-containing polyimides were obtained, which was group 1.

### 3.3. Effects of Different Types of Dianhydride on Properties of Silicon-Containing Polyimides

According to the characterization results of Experiment 3.2, PI films had the best comprehensive performance when the ratio of DSX to ODA was 1:9. Therefore, Experiment 3.3 used the ratio above to explore the influence of different dianhydride combinations on the performance of PI film with DSX structure.

In the ratio of 1:9, ternary homopolymerization with BPDA and BTDA was carried out, respectively. The experimental process was consistent with the above, and silicon-containing polyimide films were obtained by polymerization of different dianhydrides with two kinds of diamines, which was group 2. Then, tetrad polymerizations were carried out with PMDA, BPDA and BTDA, respectively, to obtain a series of silicon-containing polyimides, which was divided into group 3. Films of group 2 and group 3 were successively characterized as above.

### 3.4. Measurements

The Fourier transform infrared (FTIR) spectra of PI films were obtained using a Nicolet 6700 Fourier Transform Infrared Spectrometer (Thermo scientific, Shanghai, China). Thermogravimetric analysis (TGA) of the polyimides was carried out with a TGA 4000 (PerkinElmer, Waltham, MA, USA) at a heating rate of 10 °C/min from 30 °C to 750 °C under N_2_ atmosphere. Differential scanning calorimetry (DSC) of the composites was tested by a DSC 8000 (PerkinElmer, Waltham, MA, USA) at a heating rate of 20 °C/min from 30 °C to 450 °C under N_2_ atmosphere. The dielectric constants and loss were determined by a Dielectric Spectrometer (Wuyi Electronic, Shanghai, China) in the range of 1 Hz to 1 MHz at 29% relative humidity.

The hygroscopicity of PI films was measured according to GB/T1033-1998. The details were as follows.

The PI films were cut into square pieces the size of 5 cm × 5 cm and dried in a 60-degree oven for 24 h. The mass m_1_ was then weighed. Then, it was soaked in deionized water for 24 h and weighed immediately after wiping the water on the surface of the film to obtain the mass m_2_. Each sample was measured three times and averaged. The hygroscopicity can be obtained by the following formula.
(m_2_ − m_1_)/m_1_ × 100%(1)

## 4. Result

### 4.1. Structural Elucidation: FTIR

As shown in Figure 3, the bending vibration peak of the C=O bond could be found at 723 cm^−1^, the stretching vibration peak of the C-N bond could be seen at 1379 cm^−1^ and the stretching vibration peak of the C=O bond could be found at 1776 cm^−1^ and 1720 cm^−1^, proving that the imide structure was formed in the polymer. The stretching vibration peak of the Si-O-Si bond could be observed at 1054 cm^−1^, which proved that the silicon-containing group of DSX was successfully introduced into the molecular structure of PI. The stretching vibration peaks of the C-H bond could be seen at 2927 cm^−1^ and 2850 cm^−1^, which represented the successful introduction of the aliphatic chain of DSX into PI molecular structures. According to the infrared image, DSX was successfully introduced into the molecular structure of PI without any influence on the synthesis of PI.

Figure 4 shows that the bending vibration peak of the C=O bond could be seen at 723 cm^−1^, the stretching vibration peak of the C-N bond could be seen at 1379 cm^−1^ and the stretching vibration peaks of the C=O bond were 1776 cm^−1^ and 1720 cm^−1^, proving that the imide structure was formed in the polymer, which represented the formation of the target PI molecular structure. The stretching vibration peak of the Si-O-Si bond could be observed at 1054 cm^−1^, proving that the silicon-containing group of DSX was successfully introduced into the molecular structure of PI.

### 4.2. Thermal Properties

Since there was no water in PI films, weight loss was directly considered as polymer decomposition. The following TGA curves and T_0.5_ (the temperature at 0.5% weight loss), T_1_ (the temperature at 1% weight loss) and T_5_ (the temperature at 5% weight loss) tables were obtained.

As seen from the curves and Table 2, the T_5_ and the glass transition temperature (Tg) of PI films in group 1 decreased with the increase in DSX in the proportion of diamine. This was because the silicon-containing monomer DSX belonged to aliphatic diamines and had no rigid structure with high-temperature stability, such as a benzene ring. Compared with PI films polymerized by pure ODA and PMDA, however, the Tg and T_5_ of PI films in group 1 were all above 300 °C and 500 °C. The thermal stability did not decrease too much and was still within the range required by 5G equipment for the thermal properties of materials. The thermal performance was best when the ratio of DSX to ODA was 1:9, Tg was 318 °C and T_5_ was 551 °C.

According to the Tg and T_5_ data listed in Table 3, the T_5_ of the BPDA group was above 530 °C, with excellent thermal performance, and the Tg of the BPDA group was 325 °C at the highest and 297 °C at the lowest, which was within the acceptable range. In the BTDA group, although Tg was above 310 °C, the highest T_5_ was only 520 °C, showing slightly poor thermal stability compared with the BPDA group. From what has been discussed above, it could be concluded that BPDA was better than BTDA in neutralizing the weakening effect of DSX on the thermal properties of PI films.

### 4.3. Mechanical Properties

When PI films are applied to 5G, more attention is paid to their flexibility, so the tensile strength and elongation of PI films will be focused on when the mechanical properties are characterized.

As seen in Table 4, with the decrease in the mole content of DSX in the diamine system, there was no obvious linear relationship between the properties of the polyimide films obtained. Although in the ratio of 1:3 to 1:9, tensile strength and elongation at break increased with the decrease in the DSX proportion, in the whole set of data, tensile strength and elongation at break of 1:9 were not the best. When the ratio was 1:13, PI films had the best mechanical properties, with a tensile strength of 84 MPa and elongation at break of 14.45%.

As seen in Table 5, when all PMDA was changed into BPDA or BTDA, the tensile strength and elongation at break were decreased, but when only 50% of PMDA was changed into BPDA or BTDA, the tensile strength improved greatly, and the elongation at break did not change much. It could be inferred that the unit molecular structure obtained by polymerization of BPDA and BTDA with DSX was stronger than that of PMDA, resulting in more brittle PI films. However, if only half of PMDA was changed into BPDA or BTDA, due to the uncertain combination of the four monomers, the PI molecule could have a variety of unit structures, and at the same time, it also increased the proportion of benzene ring in the unit molecular structure, so that the tensile strength of polyimide films significantly increased. The formula with the best mechanical properties was 50%BTDA/50%PMDA//90%ODA/10%DSX, with a tensile strength of 142 MPa and elongation at break of 7.12%.

### 4.4. Dielectric Properties and Hygroscopic

According to the curves shown in Figure 5, with the increase in frequency, the Dk of the PI film first presented a downward trend and then stabilized within a certain range after decreasing to a certain extent. This was because the chemical structure and composition of DSX are very different from ODA. When DSX and ODA were randomly arranged on both sides of PMDA in aggregation, the molecular structures of the PI were full of uncertainty, and the symmetry was greatly reduced. In addition, the molecular weight of DSX is larger than ODA, making the unit structure of the PI molar volume increase and reducing the proportion of imide groups in the unit structure. Both of the above reasons can effectively reduce the polarization degree of the PI molecule, thus reducing the Dk and Df. As shown in Figure 5, with the decrease in DSX content in the diamine system, the permittivity of PI films increased and gradually became closer to PI film polymerized by pure PMDA and ODA. It could be observed that the additional amount of DSX had a great influence on the permittivity of PI films. The characterization results showed that the PI films of group 1 all had Df in the range of 10^−3^ at 1 MHz, which could meet the requirements of applied industry.

According to the data in Table 6, in the same size and humidity environment, the water absorption rate of PI films decreased as the ratio of DSX and ODA increased. When the ratio was 1:17, the moisture absorption rate was greater than 2%. This was because the molecular structure of DSX was an aliphatic chain containing a Si-C bond and Si-O-Si bond. Compared with ODA containing a benzene ring and other rigid structures, the intermolecular force of DSX was much smaller. The introduction of DSX into the PI unit structure reduced the viscosity of the new pre-polymer, weakened the surface tension of PI films and reduced the surface energy of PI films. As for hygroscopicity, the hydrophobic ability was enhanced. When the ratio was 1:3, the water absorption of the PI film was the lowest, only 1.18%.

According to the data in Table 7, DSX could effectively reduce the Dk and Df of PI film in comparison to group 2 or group 3. However, the longitudinal comparison of group 2 and group 3 found that the Dk and Df of ternary quaternion polymerization products of BPDA were both lower than those of BTDA. Therefore, a control group was specially designed for this experiment, namely, ternary polymerization without DSX participation. The Df of BPDA/PMDA//ODA and BTDA/PMDA//ODA was similar to that of BTDA//PMDA/ODA, both of which were in the range of 0.007–0.008. However, the Dk of BPDA was greatly reduced, which meant that BPDA was more suitable for DSX polymerization aiming at enhancing dielectric properties.

As for the moisture absorption rate, the hygroscopies of the formulations in the table above were all higher than 10%DSX/90%ODA//PMDA because in this group of PI films, the favorable structural effect of DSX on reducing hygroscopies was weakened by the decrease in the proportion of DSX to the overall structure. However, the hygroscopicity of PI film obtained by BPDA//90%ODA/10%DSX, 50%BPDA/50%PMDA//90%ODA/10%DSX and 50%BTDA/50%PMDA//90%ODA/10%DSX increased by a relatively small degree. The moisture absorption rate of all PI films was below 2.0%, which meets the requirements of use.

## 5. Conclusions

In conclusion, BPDA is more suitable than BTDA to introduce a DSX structure diamine polymerization system, which can greatly enhance thermal performance. However, the effect of polymerization of BPDA and a diamine system alone on dielectric properties cannot be ignored, so it is mixed with PMDA in a certain mole ratio. Then, random copolymerization with a diamine system is conducted, which can introduce BPDA’s strong thermal performance structure into the pre-polymerization system while retaining the conventional PMDA structure, playing a role in neutralizing properties. In Experiment 3.2, when the ratio of diamines was 1:9, PI films had the best comprehensive performance. The performance data were as follows: the Vitrification temperature was about 320 °C, T_5_ was 551 °C, water absorption was 1.56%, Dk was 2.35, Df was 0.007, tensile strength was 70 MPa and elongation at break was 10.2%. In Experiment 3.3, the PI polymerization formula with the best overall performance was DSX:ODA = 1:9, BPDA:PMDA = 1:1, and the properties of the films were as follows: the Dk was 2.37 (1 MHz), Df was 0.00674 (1 MHz), Tg was 325 °C, T_5_ was 530 °C, moisture absorption was 1.85%, tensile strength was 102 MPa and elongation at break was 9.21%. Compared with the results of the best proportion in Experiment 3.2, the mechanical properties were enhanced, and the excellent dielectric properties, thermal properties and hydrophobic properties were retained.

The above results showed that using the existing dianhydride monomer, DSX and ODA for mixed polymerization could reduce the Dk of PI to 2.1 and Df to 0.0068. Moreover, due to the mature industrial technology of the existing dianhydride diamine monomer, the high purity (even more than 99%) of the monomers could be guaranteed without additional cost, and the preparation process was easy to operate. These advantages make the formula obtained in this study a good fit for large-scale industrial production and could promote the industrialization of the excellent dielectric property PI.

## Figures and Tables

**Figure 1 materials-15-02755-f001:**
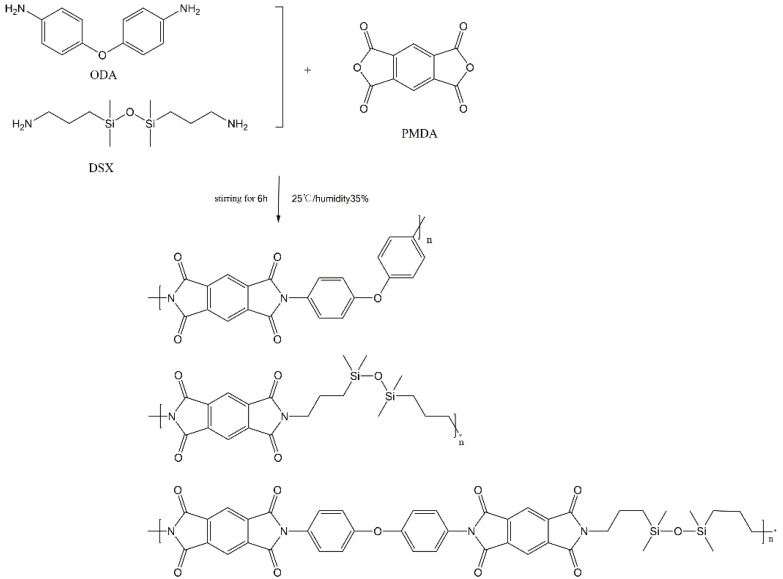
Synthesis path of polyimide (PI) films in group 1.

**Figure 2 materials-15-02755-f002:**
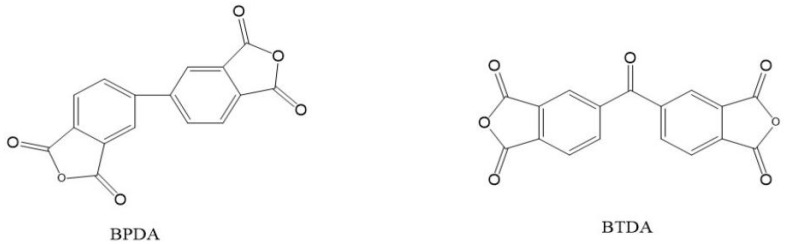
Main monomer structures in group 2 and group 3.

**Figure 3 materials-15-02755-f003:**
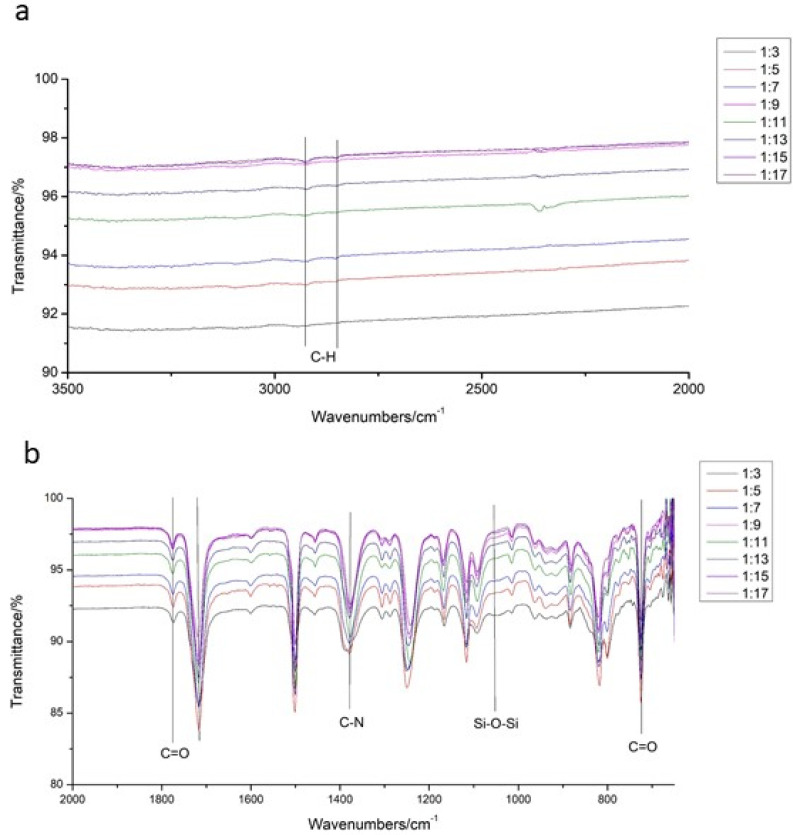
Infrared spectrum of PI films in group 1. (**a**) range of the wavenubers of bond C−H. (**b**) range of the wavenubers of other bonds.

**Figure 4 materials-15-02755-f004:**
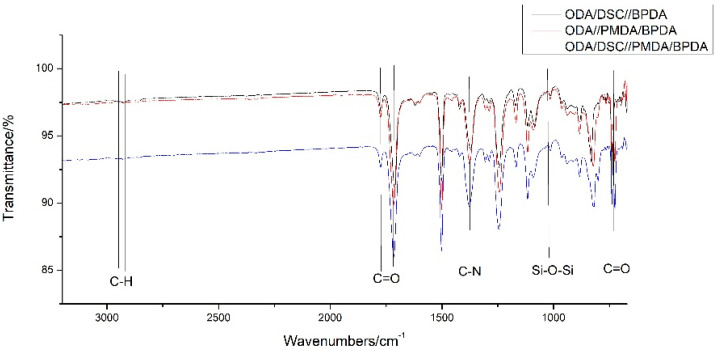
Infrared spectrograms of PI films in groups 2 and 3.

**Figure 5 materials-15-02755-f005:**
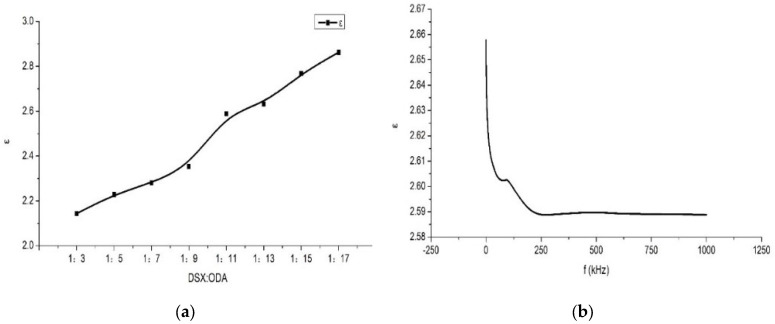
Permittivity curves of group 1, and ε represents the dielectric constant. (**a**) At 1 MHz, the change curve of dielectric constant with DSX/ODA ratio. (**b**) When the ratio of diamines was 1:11, change curve of the dielectric constant (Dk) with frequency.

**Table 1 materials-15-02755-t001:** Simulation results of the polarizability of PI films composed of each monomer group.

Monomers	Polarization Result 1/Angstrom^3^	Polarization Result 2/cm^3^	Polarization Result 3/a.u.
ODA//PMDA	41.06345	410.63453	277.11015
DSX//PMDA	45.34335	453.43347	305.99233
ODA/DSX//PMDA	85.09603	850.96031	574.25697
ODA/DSX//BPDA	94.59875	945.98749	638.38455
ODA/DSX//BTDA	105.74955	1057.49549	713.63394
ODA/DSX//BTDA/PMDA	104.44233	1044.42329	704.81237
ODA/DSX//BPDA/PMDA	95.78440	957.84398	646.38571

**Table 2 materials-15-02755-t002:** Thermal data of PI films in group 1.

DSX:ODA	T_0.5_/°C	T_1_/°C	T_5_/°C	T_g_/°C
1:3	462	473	500	306
1:5	289	442	522	310
1:7	306	468	532	315
1:9	504	512	551	318
1:11	426	474	543	319
1:13	485	507	562	318
1:15	473	504	565	322
1:17	359	449	545	327

**Table 3 materials-15-02755-t003:** Thermal properties of PI films in groups 2 and 3.

Monomers	Tg/°C	T_0.5_/°C	T_1_/°C	T_5_/°C
BPDA//90%ODA/10%DSX	297	463	477	541
50%BPDA/50%PMDA//ODA	315	450	492	570
50%BPDA/50%PMDA//90%ODA/10%DSX	325	464	480	529
BTDA//90%ODA/10%DSX	311	456	472	520
50%BTDA/50%PMDA//ODA	321	358	400	502
50%BTDA/50%PMDA//90%ODA/10%DSX	317	414	460	517

**Table 4 materials-15-02755-t004:** Mechanical properties of PI films in group 1.

DSX:ODA	Tensile Strength/MPa	Elongation at Break/%
1:3	44	4.32
1:5	56	5.91
1:7	62	7.05
1:9	70	10.2
1:11	49	6.75
1:13	84	14.45
1:15	65	8.73
1:17	83	10.23

**Table 5 materials-15-02755-t005:** Mechanical properties of PI films in groups 2 and 3.

Monomers	Tensile Strength/MPa	Elongation at Break/%
BPDA//90%ODA/10%DSX	56	5.81
50%BPDA/50%PMDA//ODA	77	4.89
50%BPDA/50%PMDA//90%ODA/10%DSX	102	9.21
BTDA//90%ODA/10%DSX	54	4.23
50%BTDA/50%PMDA//ODA	64	4.82
50%BTDA/50%PMDA//90%ODA/10%DSX	142	7.12

**Table 6 materials-15-02755-t006:** At 1 MHz, the Dk and dielectric loss (Df) of 1 PI film in each proportion.

DSX:ODA	Dk/(1 MHz)	Df/(1 MHz)	Moisture Absorption Rate
1:3	2.14	0.00814	1.18%
1:5	2.23	0.00821	1.23%
1:7	2.28	0.00743	1.35%
1:9	2.35	0.00723	1.56%
1:11	2.59	0.00681	1.72%
1:13	2.63	0.0069	1.87%
1:15	2.76	0.00757	2.05%
1:17	2.86	0.00712	2.08%

**Table 7 materials-15-02755-t007:** Dielectric properties and hygroscopicity of PI films in groups 2 and 3.

Monomers	Dielectric Constant/ (1 MHz)	Dielectric Loss/(1 MHz)	Moisture Absorption Rate
BPDA//90%ODA/10%DSX	2.64	0.0053	1.72%
50%BPDA/50%PMDA//ODA	2.67	0.00772	3.85%
50%BPDA/50%PMDA//90%ODA/10%DSX	2.37	0.00574	1.85%
BTDA//90%ODA/10%DSX	2.59	0.00801	2.63%
50%BTDA/50%PMDA//ODA	3.03	0.00782	2.63%
50%BTDA/50%PMDA//90%ODA/10%DSX	2.63	0.00568	1.96%

## Data Availability

The data used to support the findings of this study are available from the corresponding author upon request.

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
