# Peer review of "Synthesis and Properties Comparison of Low Dielectric Silicon Containing Polyimides"

_materials, 2022, doi:10.3390/ma15082755_

Round 1
Reviewer 1 Report
This paper investigates properties of low dielectric silicon containing polyimides synthesized by different routes. Unfortunately, the work is very poorly written. It is full of incorrect data, misleading statements and unconvincing discussions. Below are only a few examples:
- Discussion of Figs. 3 and 4
- There is a text: “The stretching vibration peak of Si-O-Si bond could be observed at 1054cm-1 , which proved that the silicon-containing group of DSX has been successfully introduced into the molecular structure of PI. The stretching vibration peaks of C-H bond could be seen at 2927cm-1 and 2850cm-1 , which represented the successful introduction of the aliphatic chain of DSX into PI molecular structures.” - I could not see any peak at 1054 cm-1, also the peaks at 2927 and 2850 cm-1 are hardly seen;
- Many of the peaks are not identified at all;
- It seems that Fig. 4a and 4b are the same
- Discussion of Fig. 4 repeats a part of discussion of Fig.3. Existence of Si-bonds in samples presented in Fig. 4 is not commented at all.
- Thermal properties
- The only figure in this section (Fig.5) is not commented at all;
- There is a text: “As can be seen from the curves and Table 5, both the glass transition temperature and T5 of PI films in group 1 decreased with the increase of DSX in the proportion of diamine.” – There is not any curves which show these results and the number of the table is mistaken (it should be Table 2). The results presented in the table do not support this statement for Tg. When comparing results in Tables 2 and 3, it is seen that for group 1 (Table 2) Tg>T5 (except for 1:5 and 1:7 samples) and for group 2 and 3 samples (Table 3) Tg<T5. This is very strange and is not commented at all.
- The text between Fig.5 and Table 3 repeats the text above Fig. 5.
- Discussion of Table 3 is very unconvincing.
- Conclusion
- There is a text; “In Experiment 3.2, when the ratio of diamines was 1:9, PI films had the best comprehensive performance. The performance data were as follows: Vitrification temperature was about 320℃ , T5 was 551℃, water absorption was 1.56%, Dk was 2.35, Df was 0.007”. – From Table 2 for this sample Tg is 504℃ and T5 is 318℃. With respect to dielectric properties the sample 1:7 is better with water absorption of 1.35%, Dk of 2.28, Df of 0.007.
Author Response
Thank you very much for you kind and constructive comments.
We have tried our best to revise the manuscript according to your comments,and our response to the questions are shown in the cover letter below.
We sincerely hope that this revised manuscript has addressed all your comments.And once again, thank you very much for you comments and suggestions.

Reviewer 2 Report
This manuscript provided a comprehensive comparison of various silicon-containing polyimides based on simulation and experimental results. More specifically, the effects of dissimilar monomer combinations on the overall dielectric properties of the polyimide materials are examined. Through simulation on Material Studio, experimental studies on the thermal, mechanical, and dielectric properties of the samples, the authors drew the important conclusion that the dielectric constant generally decreases as the silicon content increases, and that BPDA appears to be a more ideal material. Considering that the study is well-founded with a plethora of evidence and that these may be conducive to the field’s growth, this manuscript is recommended for publication in Materials with minor revisions.
While sufficient evidence is supplied to back up the author’s claim that BPDA is more suitable than BTDA to introduce DSX structure diamine polymerization system, more insights should be added to elucidate the significance of this finding and what these results may entail for the future design of low Dk and Df materials. This is especially crucial for expanding the scope of this manuscript. Minor typos should also be addressed, such as the subtitle “Mechanical propertiess” on page 7 should be corrected to “Mechanical properties”. On page 8, “permittivitiy” should be corrected to “permittivity”. It is recommended that the author carefully proofread the manuscript again to ensure that no typos are present.
Author Response
Point 1: More insights should be added to elucidate the significance of this finding and what these results may entail for the future design of low Dk and Df materials.
Response 1:We added some insights of significance and influence on PI industrialization as follows, ‘The above results showed that using the existing dianhydride monomer, DSX and ODA for mixed polymerization could reduce the Dk of PI to 2.1 and Df to 0.0068. Moreover, due to the mature industrial technology of the existing dianhydride diamine monomer, the high purity(even more than 99%) of the monomers could be guaranteed without additional cost. And the preparation process was easy to operate, these advantages made the formula obtained in this study a good fit of industrial large-scale production, and could promote the process of excellent dielectric property PI industrialization.’
Point 2: Minor typos should also be addressed, such as the subtitle “Mechanical propertiess” on page 7 should be corrected to “Mechanical properties”. On page 8, “permittivitiy” should be corrected to “permittivity”. It is recommended that the author carefully proofread the manuscript again to ensure that no typos are present.
Response 2:We proofread the whole article carefully and corrected some typos.
Reviewer 3 Report
In this work, Chen et al. propose a method to synthetize Si-containing polyimides with good dielectric properties (low dielectric constant and low loss) for new generation TLC.
Results are interesting and properly discussed, the used methodology is correct and sufficiently rigorous, even if some points should be clarified, as I will describe in details in my list of comments to authors.
By the way, the paper is well-organized and the topic is of timely interest. I think that it should deserve publication in Materials journal after minor revision.
Herewith the list of comments and issues to be addressed before publication:
- Authors should clearly highlight the novelty of their work. Is it the first time that dielectric properties of Si-containing PIs are improved with the method they propose (i.e. DSX and ODA mixing, followed by polymerization with PMDA, BTDA, and BPDA and tetrad co-polymerization)? A few lines should be added to the introductory section explaining this.
- In the introductory section, authors write that Dk is 2.35. It’s an interesting result, that should be better highlighted in my opinion, and contextualized by citing some references reporting Dk values recently obtained on Si-containing PIs.
- Authors should clearly explain what the values shown in Table 1 do represent. They should define polarizability, explaining to what they refer when they write “Polarization result1, 2 and 3” giving values in Å^3, cm^3 and a.u., respectively.
- When describing the experiment, the stirring speed should be specified (Page 3, line 118).
- When showing and commenting the result of thermal properties (Tables 2 and 3) authors should specify the range required by 5G equipment, otherwise it’s difficult to understand the rationale of their choice. For instance, they write that the best result was “when the ratio of DSX to ODA was 1:9, with Tg=318°C and T5=551°C”. As far as I can see from Table 1, there are some Tg values higher than 318°C (e.g. 327°C), as well as some T5 values higher than 551°C (e.g. 562°C). So why did they choose exactly 318°C and 551°C ? Specs and requirements on Tg and T5 in terms of desired ranges and optimal values should be clearly defined in the paper.
- When commenting on Ts (line 185) authors write that T5 decreased “with the increase of DSX in the proportion of diamine”. No, it’s the opposite. T5 increases indeed with decreasing DSX content, ranging from 500°C when the ratio is 1:3 to 565°C when the ratio is 1:15. Check and correct.
- When commenting on Tg (line 186) authors state that “the glass transition temperature represented no trend”. Well, I do not agree. It appears clear that Tg increases with decreasing proportion of DSX. It is indeed 306°C when DSX:ODA is 1:3 and 327°C when DSX:ODA is 1:17. Please check and correct.
- When discussing the mechanical properties, authors write that the best choice was 50%BTDA/50%PMDA//90%ODA/10%DSX with tensile strength of 142 MPa and elongation at break of 7.12%. Indeed, the tensile strength is the highest among those reported in Tables 4 and 5, but there are fairly better elongation at break values (up to 14.45%). So they should specify the reason why tensile strength is a better figure of merit to be considered when choosing the formula with the best mechanical properties.
Minor comments:
- Page 1, line 22. T5 should be defined, because it’s the first time that it appears in the paper.
- Page 1, line 40. Replace “silicone” with “silicon”.
- Page 4, line 151. Replace “degree” with “°C”
- Figure 4. Add a text box “DSX:ODA” near the plot legend.
- Page 6, line 185. It’s Table 2 and not Table 5. Check and correct.
- At the beginning of Section 4.4 (line 231) authors write “According to the curves”. What curves? Those reported in Fig. 6b? Please specify, because it’s not clear.
- Figure 6. Enlarge both values and symbols on the axes. Replace “ε”with “Dk” for consistency with the symbol used in the paper for the dielectric constant.
- The caption of Table 7 is wrong. The values shown are the dielectric properties, not the thermal ones. Moreover, symbols “Dk” and “Df” should be used in the table columns for consistency with Table 6.
- Page 9, line 265. Authors refer to a “longitudinal” comparison of group 2 and 3 results. What do they mean with “longitudinal”? It should be clarified.
Author Response
Thank you very much for your comments.
Responses are given in the following word file. Thank you very much for your comments.

Round 2
Reviewer 1 Report
Most of my remarks are not answered, e.g., the absence of stretching vibration peak of Si-O-Si bond; the same figs. 4a and 4b - I could not believe authors explanation. This is one and the same figure. Even the labels are the same. The existence of Fig.5, which is neither mentioned nor commented in the text.
Author Response
Responses to the suggestions are listed in the document below.
